# Effects of Nanoparticles on Algae: Adsorption, Distribution, Ecotoxicity and Fate

**Feng Wang [1,*], Wen Guan [1], Ling Xu [1], Zhongyang Ding [2,*], Haile Ma [1], Anzhou Ma [3] and Norman Terry [4]**

[1] School of Food and Biological Engineering, Jiangsu University, Zhenjiang 212013, China; ghwconan@163.com (W.G.); lxu@ujs.edu.cn (L.X.); mhl@ujs.edu.cn (H.M.)

[2] Key Laboratory of Carbohydrate Chemistry and Biotechnology, Ministry of Education, School of Biotechnology, Jiangnan University, Wuxi 214122, China

[3] Research Center for Eco-Environmental Sciences, Chinese Academy of Sciences, Beijing 100085, China; azma@rcees.ac.cn

[4] Department of Plant and Microbial Biology, University of California, Berkeley, CA 94720, USA; nterry@berkeley.edu

[*] Correspondence: fengwang@ujs.edu.cn (F.W.); bioding@163.com (Z.D.);
Tel.: +86-511-8878-0201 (F.W.); +86-510-8591-8221 (Z.D.)

**Abstract:** With the rapid development of nanotechnology and widespread use of nanoproducts, the ecotoxicity of nanoparticles (NPs) and their potential hazards to the environment have aroused great concern. Nanoparticles have increasingly been released into aquatic environments through various means, accumulating in aquatic organisms through food chains and leading to toxic effects on aquatic organisms. Nanoparticles are mainly classified into nano-metal, nano-oxide, carbon nanomaterials and quantum dots according to their components. Different NPs may have different levels of toxicity and effects on various aquatic organisms. In this paper, algae are used as model organisms to review the adsorption and distribution of NPs to algal cells, as well as the ecotoxicity of NPs on algae and fate in a water environment, systematically. Meanwhile, the toxic effects of NPs on algae are discussed with emphasis on three aspect effects on the cell membrane, cell metabolism and the photosynthesis system. Furthermore, suggestions and prospects are provided for future studies in this area.

**Keywords:** nanoparticles; algae; ecotoxicity; adsorption; distribution; food chain

## 1. Introduction

With the wide application of nanoparticles (NPs) in different areas, the development of nanotechnology has significantly increased. Nanoparticles are defined as particles with at least one dimension in the range of 1 to 100 nm [1]. NPs can be divided into five categories based on their chemical composition: nano-metal, nano-oxide, carbon nanomaterials, quantum dots and other NPs such as organic polymers [2]. These NPs have many applications in the fields of food packaging, textiles, optoelectronics, biomedicine, cosmetics, energy and catalysis due to their unique capabilities such as their mechanical properties, contact reactivity, optical properties and electrical conductivity [3,4]. NPs have been used in a variety of common products such as household appliances, cleaning agents, clothes, tableware and children's toys [5]. Currently, the most common nanomaterials include silver, fullerene and carbon nanotubes, zinc oxide, silica and titanium dioxide [6]. For example, Ag NP-based disinfectants have attracted much attention due to their practical applications in daily life [7].

Nanoparticles have been present on Earth from the beginning of its existence, for example, in volcanic dust, water, soil and sediment [8]. With the advent of the industrial revolution, extensive burning of a large amount of fossil fuels resulted in a significant increase in the number of NPs and has become a potential risk to the environment [9]. Currently, NPs are employed in a wide variety of industries [10]. NPs will enter into the environment during production, transportation, consumption and disposal, and eventually deposit onto bodies of water and soils [11]. In fact, many large-scale chemical manufacturers producing NPs discharge effluent into the ocean or rivers. A large number of NPs are discharged into the water body each year [8]. In a water environment, NPs pose a risk to the aquatic ecosystem and human health through a series of complex processes such as adsorption, desorption, suspension, sedimentation, and acting with minerals, organic matter, biofilm and other complex aspects of the water environment [6,12]. The use of consumer products containing NPs will expose humans to the risks associated with NPs, especially those that are in contact with the human body, such as sunscreens and cosmetics. Sunscreen contains titanium dioxide ($TiO_2$) and zinc oxide (ZnO) nanoparticles, which produce free radicals in the light that not only destroy the formulation of the sunscreen, but also destroy the biomolecules, so as to bring a risk to the consumers [13]. $TiO_2$ nanoparticles were also used as antibacterial agents and colorants in paints and food packaging [14,15], which were likely to then migrate into the environment or food sources and cause potential harm to organisms. In addition, nanoparticles in food can penetrate into the digestive tract lymphatic vessels, and compared with other large particles, can be more easily distributed into other tissues and organs [16].

Organisms of different trophic levels are usually selected to assess the ecotoxicity of NPs, including primary producers (algae), consumers (mammals, crustaceans) and decomposers (microbes) [17]. Among them, algae are the most widely used to evaluate the ecotoxicity of NPs. In aquatic ecosystems, algae lie at the lowest trophic level and are the basis of many food webs [18,19]. Algae play an important role in aquatic ecosystems and are the main producers of aquatic food chains [20]. They do not only provide oxygen and food for other aquatic organisms through their own photosynthesis, but also contribute to the purification of water [21]. Algae have been used as a bio-indicator of pollutants due to their high bioaccumulation ability [20]. Because algae cells are sensitive to many poisons and have the advantages of a short growth period, can be easily isolated and cultured, can be directly observed and show the symptoms of poisoning at the cellular level [22], many countries use algae for chemical risk assessment and have established several standardized test methods. Therefore, algae can be used as a model organism to study the biological toxicity of nanoparticles. At present, many studies have been conducted in this area. Most of the research has focused upon the toxic effects of nanoparticles on algae. The results show that nanoparticles had effects on the growth condition, chlorophyll content, protein content and enzyme activity of algae. The toxic effects were related to the morphology, size, chemical composition, concentration, solubility and dispersion of the nanoparticles, which was also dependent on the cell structure and physiological and biochemical characteristics of tested algae [23–25].

Some contents on the ecotoxicity of NPs have been summarized in published review papers. These reviews mainly summarized the behavior and fate of NPs in the environment, including the interaction of NPs with the organic matter in the environment, as well as the interaction between NPs and organisms, and highlight the adverse effects of NPs on organisms such as algae, plants and fungi [26,27]. The toxicity of NPs to microalgae and their behavior in the environment were also mentioned in some reviews [23,28]. Bioassay methods for nanotoxicology of microalgae were discussed and recommendations for further research were provided [29]. In this paper, we focused on the effect of NPs to algae. For this purpose, this review will systematically summarize and discuss the adsorption and distribution of NPs in algal cells, the ecotoxic effects of NPs on algae and affecting factors, as well as their fate in the aquatic food chain.

## 2. Adsorption of NPs by Algae

As a model organism for studying the biological toxic effects of NPs, it is necessary to investigate the adsorption capacity of algae to NPs. The adsorption capacity of algae to NPs has been tested in some research. The results are summarized in Table 1. For Ag NPs, the accumulated Ag

concentrations in the algae *Chlorella vulgaris* reached 1200–3300 µg/g dry weight after 4 h of exposure to 2 mg/L [30], and the accumulated Ag concentrations in the algae *Raphidocelis subcapitata* reached 45.0 µg/g dry weight after an exposure of 24 h to 15 µg/L and 93.7 µg/g dry weight after exposure of 24 h to 30 µg/L [31]. These results indicated that algae can absorb a large amount of NPs. However, the absorption capacities of different species of algae to different NPs varied in a wide range. Meanwhile, the absorption capacity was also related to other reaction parameters, such as the concentration of NP suspension and the time of algal exposure to NPs.

**Table 1.** Adsorption capacity of algae to nanoparticles (NPs).

| Nanoparticles | Algae | Exposure Concentration | Exposure Duration | Concentration of NPs in Algae (Dry Weight) | Reference |
|---|---|---|---|---|---|
| Silver (Ag) | *Chlorella vulgaris* | 2 mg/L | 4 h | 1200–3300 µg/g | [30] |
| | *Raphidocelis subcapitata* | 15 µg/L 30 µg/L | 24 h | 45.0 µg/g [1] 93.7 µg/g [2] | [31] |
| Titanium oxide (TiO$_2$) | *Scenedesmus obliquus* | 10 mg/L | 72 h | 59.51 ± 2.16 µg/g | [32] |
| | *Scenedesmus acutus* | 0.3 mg/L 1.2 mg/L 4.8 mg/L | 72 h | 490 ± 143 µg/g 1714 ± 488 µg/g 5701 ± 3163 µg/g | [33] |
| Carbon nanotubes (CNT) | *Desmodesmus subspicatus* | 1 mg/L | 24 h 48 h 72 h | $(1.30 ± 0.42) × 10^3$ µg/g $(2.11 ± 1.76) × 10^3$ µg/g $(4.98 ± 1.63) × 10^3$ µg/g | [34] |

[1] The percentage of overall NPs absorbed by algae cells was 21%. [2] The percentage of overall NPs absorbed by algae cells was 31%.

In the study of algal adsorption dynamics, two kinetic models, the pseudo-first order model and the pseudo-second order model, are often used to analyze the kinetics of algal adsorption to NPs. Two isotherm models, the Langmuir and Freundlich models, are widely used to fit the adsorption isotherm experimental data [35,36]. In general, the studies of adsorption kinetic and isotherm models show that the adsorption of algae to NPs is best fitted to a pseudo-second order kinetic model and Langmuir isotherm model [35,37,38]. The initial concentration of NPs in water influenced the adsorption of algae to NPs. With an increase in initial concentration, the adsorption capacity of algae to NPs increased [37]. The adsorption process was very fast at the initial stage, but afterwards the rate of adsorption decreased slowly until it reached saturation [37]. The amount of NPs adsorbed by algae from the water varied significantly with the change of solution pH. The relatively strong acidic pH ranges can thus enhance the adsorption of algae to NPs [36]. At the same time, the absorption of algae to NPs was affected by the shape, biomass dose and physicochemical properties of algae [35].

For the uptake of NPs into algae, NPs need to move across two barriers (i.e., cell walls and plasma membranes) [25]. For algae, the relatively thick and tough cell walls are generally considered as the first barrier to prevent internalization of NPs. The algae cell walls are semipermeable, and usually porous in their structure [26]. The diameter of these pores is in the range of 5–20 nm [39,40]. Generally, only the NPs in the size smaller than the maximal pore size can easily pass through the cell wall, while limiting the passage of larger molecules [30]. The main component of the cell wall in algal cells is cellulose, which also usually contains glycoproteins and polysaccharides [26]. These components can act as binding sites to promote the adsorption of NPs by algae [41,42]. Once the NPs pass through the cell wall, NPs will encounter the second barrier—the plasma membrane. Endocytosis and passive diffusion are considered to be the main pathway for NPs to cross the bilayer lipid membrane [43,44]. In addition, the permeability of the cell wall may change during cell cycling, with the newly synthesized cell wall being more permeable to NPs, thereby increasing the uptake of NPs by algal cells [30,45].

### 3. Distribution of NPs in Algal Cells

When NPs enter the algal cells, they can destroy the cell wall and cell membrane, and will be deposited in the space between the cell wall and the plasma membrane (i.e., the periplasmic space) [19,46]. NPs can enter the cytoplasm and make contact with some organelles of cellular structures, including chloroplasts, vacuoles, endoplasmic reticulum, Golgi apparatus and mitochondria, and will significantly damage or alter their function and structure [20,47]. It has been observed that NPs entering cells will damage the structure of chloroplasts. In the chloroplasts, the chloroplast membrane was damaged, and the grana lamellae of the thylakoids was also be destroyed, creating a messy state [46,48]. CuO NPs were observed to be clearly deposited in the vacuoles of algal cells [47]. Heavy metals can accumulate in the vacuoles of *Chlamydomonas acidophila* cells, likely as a detoxification mechanism [49]. One study has shown that endoplasmic reticulum swelling was clearly observed in algal cells exposed to high doses of single-walled nanotubes [50]. NPs can also cause mitochondrial dysfunction, which may affect the metabolic process of algal cells [47]. In addition, NPs can enter the nucleus, leading to abnormal nuclear effects [20,51]. It was found that nuclear chromatin clumped and became condensed against the nuclear membrane [46]. This can cause DNA damage and thereby inhibit the process of cell division. Internal degradation of the nucleus and chloroplasts occurred in algal cells treated by ZnO NPs [20]. There was also NPs aggregation in other cytoplasmic spaces [19]. In short, after entering the algal cell, NPs will be spread to various parts of the cell. It will not only destroy the cellular structure, but also influence the metabolism and reproductive function of the algal cell. However, very few studies have described the transport mechanism of NPs in algal cells, which remains to be studied. The possible distribution and potential toxic outcomes of NPs in algal cells are shown in Figure 1.

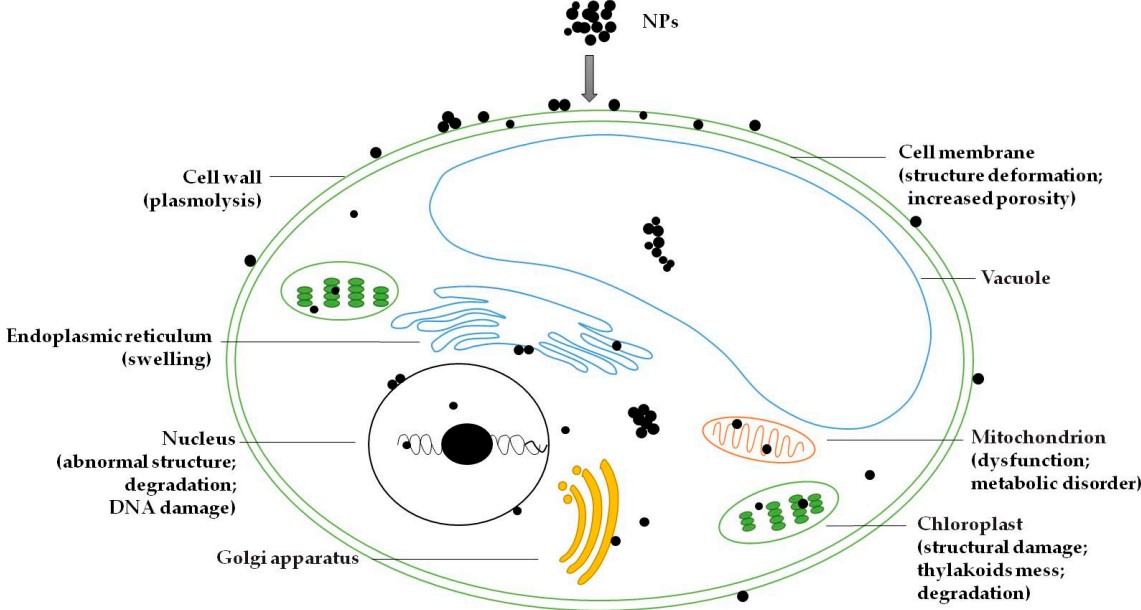

**Figure 1.** Possible distribution and potential toxic outcomes of NPs in algal cell.

### 4. Ecotoxic Effects of NPs on Algae

#### 4.1. Effect of NPs on the Cell Membrane

NPs have a high reactivity in the liquid phase, and the toxicity of the NPs that aggregate through the cell wall to the cytoskeleton structure (mainly referring to the cell membrane) is significant. The interactions of the NPs with algae caused damage to the cell membrane and the release of lactate dehydrogenase into the test solution, which may be one of the toxicity mechanisms inducing cell death [20]. NPs mainly cause toxicity to the cell membrane by the oxidative stress reaction, resulting in membrane damage and the decreasing of membrane integrity [45]. In a study of the toxicity of $TiO_2$

NPs to *Anabaena variabilis*, the exposure of algae to NPs brought an increase in the content of reactive oxygen species (ROS) in the cell membrane and the production of membrane protein crystals. At the same time, the cell membrane was abnormal or even damaged [52]. In the case of cytotoxicity of NiO NPs on *Chlorella* sp., algae cells appeared on the cell wall separation and cell membrane rupture in the presence of NiO NPs [53]. One study has shown that NPs may cause membrane changes and increase its porosity, thereby further facilitating NPs to enter cells [54]. It was deduced that contact with NPs induced the formation of new pores in the membrane through lipid peroxidation mechanisms, making it more permeable and less selective [51]. Furthermore, NPs can also accumulate in the cell membrane and lead to cell wall depression, which causes changes in cell membrane permeability, until cell apoptosis occurs [20]. Nanoparticles have been found to increase lipid peroxidation in cell membranes, resulting in membrane structure deformation [55]. In the cytotoxicity study of $TiO_2$ NPs toward *Scenedesmus obliquus*, plasmolysis and dentate cell membrane were observed. In addition, the cell membrane thickened with the presence of $TiO_2$ NPs, which can be attributed to the protective nature of the cells, preventing the internalization of NPs [22].

### 4.2. Effect of NPs on Intracellular Substrates

NPs can induce the production of intracellular reactive oxygen species (ROS), such as superoxide radicals ($O_2^-$), singlet oxygen ($^1O_2$), hydroxyl radicals (HO·) and hydrogen peroxide ($H_2O_2$) [56], which induce oxidative stress in cells, resulting in oxidative-reduction imbalance in an organism [46,47]. Oxidative stress refers to the body in a variety of harmful stimuli, the body of high activity of free radicals and the imbalance of the oxidation system and antioxidant system, which eventually leads to tissue damage [20]. It has been proven that the production of ROS can damage intracellular lipid, carbohydrate, protein, DNA and other biological macromolecules, leading to inflammation and oxidative stress [16]. ROS can interact with the side chain of polyunsaturated fatty acids and nucleic acids and other macromolecules, resulting in lipid peroxidation [45]. Lipid peroxidation is regarded as the most severe form of damage resulting from oxidative stress as it can lead to changes in the cell membrane system, which in turn disrupts the cellular function of the organism [51]. Oxidization of nucleic acids by ROS would also cause mutagenesis [57]. ROS can result in the formation of many lipid decomposition products. Some of these decomposition products are harmless and others may cause disordered cell metabolism and dysfunction [58,59]. ZnO NPs facilitated the formation of phosphate granules, starch pyrenoid complexes and lipid droplets in *Scenedesmus obliquus* cells, which was observed by transmission electron microscopy [20]. Similarly, ZnO NPs were also proven to induce phosphate granule formation in *Chlorella pyrenoidosa* and *S. obliquus* by Zhou et al. [60]. To control ROS level in cells, cells have evolved an ROS scavenging system of enzymatic and nonenzymatic antioxidants, such as ascorbic acid (AsA), glutathione, tocopherol, carotenoid, catalase (CAT), superoxide dismutase (SOD), peroxidase (POD) and ascorbate peroxidase (APX) [61,62]. It was suggested that ROS production led to enhanced production of antioxidants and increased activity of antioxidant enzymes for the activation of the complex cellular defense mechanisms [51]. These antioxidants were able to neutralize free radicals or their toxic effects, acting at different steps. The intervention of NPs has an impact on the protein and enzyme activity [51]. Antioxidant enzymes such as superoxide dismutase (SOD), catalase (CAT), glutathione S-transferase (GST) and peroxidase (POD), among other enzymes, are mechanisms of antioxidant protection in algal cells. In the presence of less toxic NPs, the activity of these antioxidant enzymes increases to eliminate the toxicity of ROS to algal cells [63], and when the toxicity of NPs in water is too strong, the antioxidant defense system of algae cells is destroyed and the enzyme activity is significantly inhibited [45,64].

### 4.3. Effect of NPs on the Photosynthesis System

Algae is a kind of phytoplankton, and photosynthesis is an important physiological process of algae. The adsorption of NPs on the surface of algal cells results in a shading effect that affects algal photosynthesis [26]. The shading effect caused by NPs affects the light, pigment and other conditions necessary for photosynthesis, weakening the algae absorption of light and thereby inhibiting the

photosynthesis process [65–67]. However, the nanotoxicity caused by the shading effect is hard to be quantitatively determined separately because the attachment could also lead to other forms of toxicity (e.g., physical membrane damage) [47]. The chloroplast membranes that are rich in polyunsaturated fatty acids are potential targets of lipid peroxidation, which may cause damage to photosynthesis sites [57]. *Chlorella* sp. showed sensitivity to $TiO_2$ NPs and structural damage was observed in the nucleus and the cell membrane, as well as in the chloroplasts [68]. In a toxicity test of $Al_2O_3$ NPs to *Chlorella* sp. and *Scenedesmus* sp., it was found that $Al_2O_3$ NPs could reduce the content of chlorophyll in algal cells [69]. It was also proven that $SiO_2$ NPs affect the content of the photosynthetic pigment of *Scenedesmus* sp., in which chlorophyll a and b content decreased while carotenoid content was not affected [70]. Similarly, chlorophyll a content and photosynthetic efficiency in *Scenedesmus bijugus* decreased as a function of the $TiO_2$ NPs concentration [71]. Ag NPs significantly reduced chlorophyll content and inhibited the growth of the green algae *Chlorella vulgaris* [72]. The change of photosynthetic pigment content directly influenced the photosynthesis of algal cells [51,65]. One study has shown that NPs changed the photosynthetic rate and respiration rate of algae, resulting in metabolic disturbance [46]. CuO NPs were reported to reduce PSII capacity in converting light energy in photosynthetic electron transport [65]. Table 2 summarizes the different toxic effects of some NPs on algae.

**Table 2.** Toxic effects of nanoparticles on algae.

| Nanoparticles | Diameter (nm) | Tested algae | Toxic Effects | References |
|---|---|---|---|---|
| Silver (Ag) | 20–50 | *Chlamydomonas reinhardtii* *Pseudokirchneriella subcapitata* | $EC_{50}$: 9.9 µg/L (96 h) Affecting photosynthesis, inhibition in cell growth | [19,25] |
| Copper oxide (CuO) | 30–40 | *Chlamydomonas reinhardtii* *Chlorella pyrenoidosa* | $EC_{50}$: 45.7 mg/L (72 h) DNA damage | [47,51] |
| Titanium oxide ($TiO_2$) | 20–35 | *Pseudokirchneriella subcapitata* *Desmodesmus subspicatus* | $EC_{50}$: 10–26 mg/L (72 h) Decrease in chlorophyll a and soluble protein content, accumulation of lipid hydroperoxide | [15,73,74] |
| Zinc oxide (ZnO) | 20–30 | *Pseudokirchneriella subcapitata* *Scenedesmus obliquus* | $EC_{50}$: 0.5–1.5 mg/L (96 h) Inhibition in cell growth, destruction of cell antioxidant capacity | [20,73,75] |
| Quantum dots (QDs) | 1–10 | *Chlamydomonas reinhardtii* | $EC_{50} < 5$ mg/L (72 h) Cell aggregation, lipid peroxidation | [76,77] |
| Fullerence ($C_{60}$) | <200 | *Scenedesmus obliquus* *Chlamydomonas reinhardtii* | Increase in cell death, affecting photosynthetic and respiratory processes in cytosol | [27,39] |
| Single-walled carbon nanotubes (SWCNT) | <2 | *Pseudokirchneriella subcapitata* *Scenedesmus obliquus* | $EC_{50}$: 22.6 mg/L (96 h) Inhibition in cell growth | [27,78] |
| Multiwalled carbon nanotubes (MWCNT) | 10–20 | *Scenedesmus obliquus* *Desmodesmus subspicatus* | $EC_{50}$: 15.5 mg/L (96 h) Decrease in cell viability | [27,34] |

## 5. Factors Affecting Toxicity of NPs on Algae

### 5.1. Environmental Factors

The toxicity of NPs on algae is influenced by various factors, particularly the characteristics and properties of the aquatic environment around algae, such as water chemistry, light and water

temperature [23]. The hydrochemical conditions are important factors that influence the suspension of NPs, including dissolved natural organic matter (NOM), ionic strength and pH [8]. NOM can be adsorbed on NPs to alter the surface functional groups of NPs or form thin films, and enhance their migration and diffusion capabilities. NOM stabilized the particle size by covering the surface of NPs due to electrostatic repulsion [79]. The coating of NOM may limit the release of ions from NPs into the water [80], prevent NP aggregation [81] and reduce the toxicity of NPs on the algae [82]. The ionic strength and pH of natural water bodies can change the suspended state of NPs in water [83], which also influences the adsorption of NPs to NOM [84]. Water hardness is another important NP toxicity mitigator, promoting NP aggregation and decreasing dissolution [85]. In addition, water temperature, light and pollutant emissions will affect the toxicity of NPs. It is known that temperature directly affects aquatic ecosystem communities [86] since it is regarded as an important abiotic factor influencing the growth and production of primary producers such as algae. A higher dissolution rate of Ag NPs was obtained with an increased temperature [87]. Therefore, the toxicity of NPs to algae could be affected by temperature due to the change in the physiological status of algal cells and the existing state of NPs. Because some NPs are semiconductors with photocatalytic properties, NPs exposed to UV-light may cause a toxic effect on algae through the formation of highly reactive ROS [23]. Bhuvaneshwari et al. [20] found that NPs were more toxic to algal cells irradiated with UV-C (a high energy radiation with a wavelength less than 280 nm) than those treated in dark and visible conditions.

### 5.2. Algae Characteristics

Additionally, besides the presence of substances in the water that will affect the toxicity of NPs, the toxic effects of NPs are also different when the test algae varied. The cell wall thickness, cell volume, polysaccharide and other organic matter contained in the cell wall of different algae species can affect the biological toxicity of NPs [24]. Algae itself can secrete polysaccharide-rich cellulose and other compounds to feedback the effects of exotoxic agents, which can promote the aggregation of NPs in a water environment to form relatively large particles and reduce their access to algal cells [88]. The internalization of Ag NPs was investigated in the wild type with a regular cell wall and a cell wall free mutant of *Chlamydomonas reinhardii*, where a higher accumulation rate was achieved in the cell wall free mutant, indicating a protective role of the cell wall in limiting Ag+ uptake [89]. Comparison of the toxicity of Ag NPs to Chlorococcales and filamentous algae showed that the filamentous algae *Klebsormidium sp.* was able to uptake more NPs under the same conditions [40].

### 5.3. Surface Functional Groups and NP Types

Different nanoparticles exhibit different levels of toxicity upon the growth of algae [26]. The properties of NPs, such as particle diameter, shape and surface morphology, affect the physical stability and the performance of NPs [4]. Many studies have strengthened the fact that the inherent properties of NPs, such as chemical composition, particle size, synthesis method, aggregation state, concentration and surface chemistry, may affect the toxicity of NPs on algae [3,25,90]. The toxicity of ZnO NPs and $TiO_2$ NPs are relatively stronger due to their good photosensitivity, in which the toxicity of ZnO NPs is stronger than $TiO_2$ NPs, resulting from the dissolution of zinc ions [25].

In order to prevent agglomeration and obtain the NPs with good dispersibility, the surface modification and functionalization of the NPs are often carried out by means of surface coating in the preparation process [90]. The change of the chemical properties of the NPs will affect the toxicity of NPs in the water environment. The study on the bioaccumulation kinetics of three different coated silver NPs showed that the coated NPs could lead to an increase in algal cell membrane permeability compared with the bare Ag NPs, where the toxic effects of different coated Ag NPs on algae were also different [30]. It was proven that polymer-coated CuO NPs were more toxic to the green algae *Chlamydomonas reinhardtii* than to that of the uncoated CuO NPs [65]. However, NPs can become less toxic after the coating process. The toxicity of $TiO_2$ coated with humic acid to algae weakened [91]. The Al-coated $SiO_2$ NPs have a tendency to aggregate with algal cells and thus become less toxic than uncoated $SiO_2$ NPs [23]. It may be deduced that the ecotoxicity of surface modified NPs is partly dependent on the properties of the coated function group or polymers.

### 6. Fate of NPs in the Aquatic Food Chain

As a basis for aquatic food chains, algae are the source of food for various aquatic organisms and algae can promote the ingestion of NPs by organisms during feeding, thereby incorporating NPs into food chains (Figure 2) [23,30,92]. The transmission or enrichment of NPs through the aquatic food chain may lead to toxic effects on high trophic level organisms in the food chain.

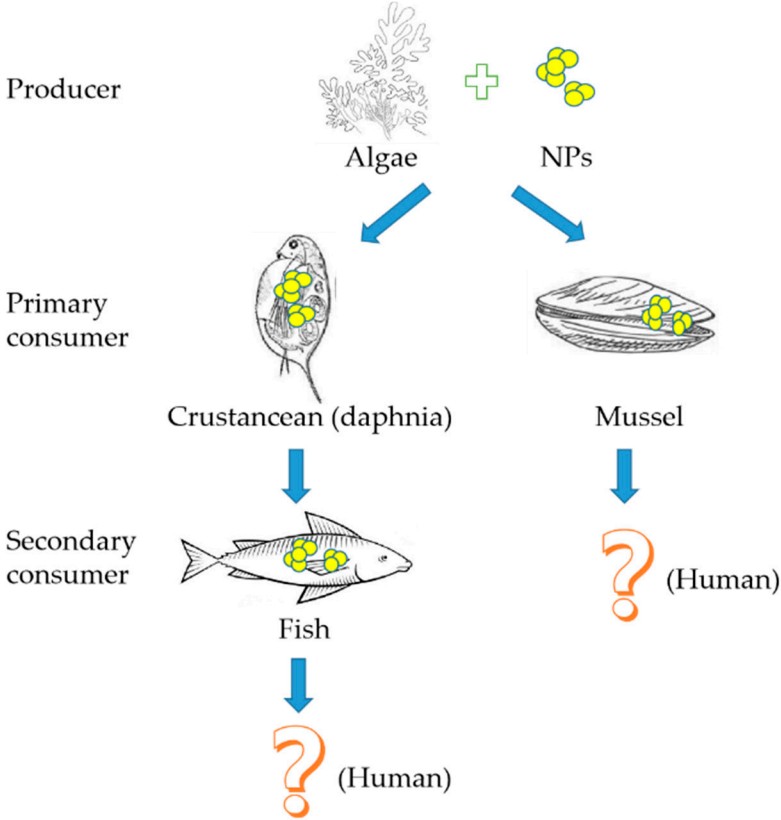

**Figure 2.** Fate of NPs in the aquatic food chain.

Since crustaceans are the major consumer of algae, there have been several studies that reported the transfer of NPs from algae to the crustacean *Daphnia* through diet. It was reported for the first time that quantum dots can be transported along the food chain from *Pseudokirchneriella subcapitata* to *Ceriodaphnia dubia* through fluorescence techniques [93]. When algae were used to feed *Daphnia magna, Daphnia magna* was found to ingest NPs from the test suspension through feeding behaviors [94]. The behavior of gold nanoparticles was studied in a model food chain consisting of a phytoplankton as food (*Ankistrodesmus falcatus*) and a zooplankton grazer (*Daphnia magna*), and a significant accumulation of Au NPs in the gut of *Daphnia* was observed [18]. Using stable isotope labeling techniques, it was revealed that carbon nanomaterials underwent complex aquatic accumulation and transfer from primary producers to secondary consumers (*Scenedesmus obliquus-Daphnia magna* food chain) [92]. Studies have shown that these NPs could accumulate in the digestive tract of *Daphnia magna* and exhibit chronic toxicity upon the growth and reproduction of *Daphnia magna* [32,95,96]; even the neonate production from adult daphnids was significantly reduced [97]. Not only can the toxic concentration of NPs cause mortality in *Daphnia*, the selectivity of daphnids to uncontaminated algal cells may also contribute to the reduced growth and reproduction of daphnids due to starvation [98,99]. It was demonstrated in a study by Dalai that the toxicity of $TiO_2$ NPs to daphnia may be partly due to the starvation of daphnids resulting from their selectivity to algal cells [22]. In this case, daphnids were exposed to $TiO_2$ NP-treated algal cells at various concentrations for a chronic exposure study. The chronic exposure resulted in a concentration dependent decrease in the body length and reproduction capacity of the daphnids [22]. The toxicity and trophic transfer of metal oxide NPs was assessed from marine microalgae *Cricosphaera elongata* to the larvae of the sea urchin *Paracentrotus*

*lividus*, where the survival rate of larvae fed with microalgae exposed to $SiO_2$ and $CeO_2$ NPs was significantly reduced, and abnormal developments with skeletal degeneration and altered rudiment growth were observed [100]. *Mytilus galloprovincialis* (Mediterranean mussel) was fed with algae previously exposed to Au NPs, and NPs were found in the gills and digestive glands of the mussels [16].

After zebrafish were fed with *Daphnia magna* containing $TiO_2$ NPs, the content of $TiO_2$ NPs in the fed zebrafish was higher than that of zebrafish exposed to $TiO_2$ NP aqueous solution alone, which is the first direct evidence for the transfer of $TiO_2$ NPs from daphnia (*Daphnia magna*) to zebrafish (*Danio rerio*) through dietary intake [101]. The toxic effects of nano-polystyrene on zebrafish delivery along the tertiary aquatic food chain of green algae–*Daphnia magna*–zebrafish were studied by Cedervall et al., and it was found that nano-polystyrene was transferred from green algae to fish and that it made a great impact on the behavior and lipid metabolism of fish [102]. NPs have toxic effects on zebrafish embryos, such as delayed hatching, reduced larvae body length and tail malformation [103]. These studies suggest that food chain enrichment is likely to be an important pathway for high trophic level bioaccumulation and enrichment of NPs. NPs can be transported through the food chain and accumulated in high trophic level organisms, and can penetrate the tissue barrier, then accumulate in the liver, kidney, spleen, muscle, stomach and intestine of high trophic level organisms [104]. Although more assessments of NPs have not been made, NPs could potentially be transferred from aquatic ecosystems to terrestrial ecosystems, and pose potential risks to humans [23].

## 7. Conclusions and Outlook

As a new kind of material, nanomaterials have been widely used in various fields even though there exists a certain degree of threat to the safety of aquatic organisms. Studies on the toxicity of NPs on algae can not only explore the toxic mechanism of NPs on algae, but also provide a theoretical basis for the safety assessment of biological toxicity of nanomaterials. For this purpose, great efforts have been made in this area and more work will be conducted in future.

The adsorption of NPs on algal cells was generally tested in the ideal system. Therefore, it is necessary that inhibitors (e.g., NOM) and accelerants (e.g., surfactants) for NPs can be investigated in detail in the adsorption process of algae, and suitable agents would need to be selected to decrease or improve the algal adsorption to NPs. This information can be used in the detoxication procedure of algae to NPs and the treatment of NP-contaminated water by algal cells.

A few studies have demonstrated the uptake of NPs into algal cells. This research mainly focused on the description of phenomena and provided some deduction about the mechanism of the uptake process. When NPs pass the cell membrane of algae by endocytosis or passive diffusion, the carrier of NPs in the cell membrane may be involved. The study on the structure and act mechanism of NP carriers can be useful for the uptake regulation of NPs by algae.

After entering the algal cell, NPs were observed on different organelles of cellular structures. Reported studies tended to explain the damage of NPs to organelles. There is still a lack of mechanism exploration for the transportation of NPs in the cytoplasm of an algal cell. Also, the quantitative distribution of NPs on different organelles is important for the evaluation of NPs ecotoxicity to algae.

The oxidative damage mechanism has been proven to be one of the major possible mechanisms for NP ecotoxicity to algae, which can cause damage to the cell wall, cell membrane and organelles. More information about the change of metabolization in an algal cell is suggested to be investigated by transcriptome analysis and metabolomic analysis. Additionally, the response and repair mechanism of algal cells to the damage done by NPs should be explored and further used to construct new algal strains that are highly resistant to NPs. Currently, high dosages of NPs are often used in NP toxicity tests on algae and the tests are generally characterized by short-term biological effects. However, the exposure dose of NPs in the natural environment is usually low, and exposure time can be much longer. Therefore, a long-term exposure study under a low dosage will provide data that are more useful in evaluating the ecotoxicity of NPs to algae. On the contrary, to relieve the damage caused by the oxidative stress in algae, the expression of an antioxidant enzyme was up-regulated and more antioxidant substrates were produced. Thus, it is possible to use the toxicity

mechanism to enhance the production of antioxidant metabolites from algal cells by utilization of NPs at a certain level.

The concentration of NPs in the water environment is low, and NPs may accumulate in organisms of high trophic level and produce significant toxic effects by the stepwise delivery or enrichment of the food chain. The present research on the transmission and biomagnification of NPs in the food chain of aquatic ecosystem is still limited. It is not clear how the environmental factors would influence the transmission of NPs in the food chain. Therefore, it is necessary to establish a more comprehensive aquatic biological system, including primary producers and consumers at different trophic levels, to investigate the transmission and biological effects of NPs along the food chain.

**Acknowledgments:** This work was financially supported by the National Natural Science Foundation of China (No. 31571822), the Natural Science Foundation of Jiangsu Province (No. BK20160493), the China Postdoctoral Science Foundation (No. 2015M571691), the Senior Talent Scientific Research Initial Funding Project of Jiangsu University (No. 15JDG17 & JDG061), the 2015 International Postdoctoral Exchange Fellowship Program of China, the 2014 Excellent Key Young Teachers Project of Jiangsu University, the Open Project of Key Laboratory of Environmental Biotechnology, CAS (No. kf2018004) and the Qing Lan Project.

**Conflicts of Interest:** The authors declare no conflict of interest.

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
