# Peer review of "Effects of Nanoparticles on Algae: Adsorption, Distribution, Ecotoxicity and Fate"

_applsci, doi:10.3390/app9081534_

Round 1
Reviewer 1 Report
Authors present review of literature on effects of nanoparticles to algae with special attention paid to effects on the cell membrane, cell metabolism and photosynthesis systems. The review is not a deep work and present very limited reviews and conclusions, it also does not present in most cases any detailed mechanisms of action of NPs on algal cells. It can be accepted after some major changes are introduced to the work.
I do not agree with statement that progress in nanotechnology has started in XXI c; already in late 80s of XX it was observable.
Please rephrase “Currently, the most common nanomaterials are silver, fullerene and carbon nanotubes, zinc oxide, silica, titanium dioxide and so on [6]” – “so on” are the most common nanomaterials.
Marine is adjective in most common usages so what is its meaning in line 50? Titanium and zinc oxides were used in paints since decades so not only using them in cosmetics creates a problem.
Line 61: remove the first sentence, it is very ordinary. Generally I would recommend using passive voice instead of active one in the entire manuscript.
Line 108: what is measure of being “gentle” in this case?
Fig. 1. presents something obvious – that NP can enter any part of algal cell.
Subchapters 4.1 till 4.3 are most interestingly presented although short.
In table 2. Please make 50 in “EC50” with lower case.
Change the title of chapter 5.
6th chapter presents some very basic phenomena, food chain is obvious route of toxification of higher organisms with food and starving/food viability studies are more detailed in the scientific literature. Please add more information on that.
Conclusions are comprehensive and properly written. There is no need to change it.
Author Response
The authors are grateful for the helpful suggestions from the reviewer. With respect to specific requests for clarification, the following revisions have been made.
Response to Reviewer 1 Comments
Authors present review of literature on effects of nanoparticles to algae with special attention paid to effects on the cell membrane, cell metabolism and photosynthesis systems. The review is not a deep work and present very limited reviews and conclusions, it also does not present in most cases any detailed mechanisms of action of NPs on algal cells. It can be accepted after some major changes are introduced to the work.
Point 1: I do not agree with statement that progress in nanotechnology has started in XXI c; already in late 80s of XX it was observable.
Response 1: Thanks for your suggestion,and this sentence has been revised.
Point 2: Please rephrase “Currently, the most common nanomaterials are silver, fullerene and carbon nanotubes, zinc oxide, silica, titanium dioxide and so on [6]” – “so on” are the most common nanomaterials.
Response 2: The sentence has been rewritten according to the reviewer’s suggestion.
Point 3: Marine is adjective in most common usages so what is its meaning in line 50? Titanium and zinc oxides were used in paints since decades so not only using them in cosmetics creates a problem.
Response 3: The word “marine” was revised to “ocean”. And also, more information about titanium oxides has been added according to the reviewer’s suggestion.
Point 4: Line 61: remove the first sentence, it is very ordinary. Generally I would recommend using passive voice instead of active one in the entire manuscript.
Response 4: The first sentence has been deleted and some sentences have been changed into passive voice in the manuscript.
Point 5: Line 108: what is measure of being “gentle” in this case?
Response 5: The sentence has been rewritten for accurate description.
Point 6: Fig. 1. presents something obvious – that NP can enter any part of algal cell.
Response 6: Figure 1 has been revised and more information was added.
Point 7: Subchapters 4.1 till 4.3 are most interestingly presented although short.
Response 7: According to the reviewer’s suggestion, more content was added in the subchapters 4.1 to 4.3.
Point 8: In table 2, Please make 50 in “EC50” with lower case.
Response 8: It has been revised.
Point 9: Change the title of chapter 5.
Response 9: The title of chapter 5 was changed to “Factors affecting toxicity of NPs to algae”.
Point 10: 6th chapter presents some very basic phenomena, food chain is obvious route of toxification of higher organisms with food and starving/food viability studies are more detailed in the scientific literature. Please add more information on that.
Response 10: Thanks for your helpful suggestion. More information has been added in chapter 6.
Point 11: Conclusions are comprehensive and properly written. There is no need to change it.
Response 11: Thank you.
Note:
1. "Track Changes" function was used during the revision of the manuscript and all the change has been marked in red.
2. More references were cited and all the references have been renumbered.
Reviewer 2 Report
This is a well put-together manuscript summarizing the effects of nanoparticles on algae. The authors did an excellent job reviewing the current literature of testing nanoparticle toxicity against algae, and concisely yet very informatively outlined the key aspects of NPs and algae interactions, including adsorption and distribution, toxicity mechanisms and outcomes, factors influencing toxicity, as well as the fate of NPs moving up the foot chain through algae. At the end of the review, the authors also pointed out the current knowledge gap and proposed several interesting research areas to further our understanding of NP toxicity.
Only minor edits are recommended to improve the overall quality of the manuscript.
· Page 3, Table 1: In the listed studies, what were the percentage of overall NPs being absorbed by algae cells?
· Page 3, lines 101-12: Can you add an example figure showing the adsorption kinetics?
· Figure 1: It would be nice to also include the potential toxic outcomes listed next to the organelles.
· Page 5, line 164: Can you list a few examples of the specific ROS produced by exposure to NPs in algae?
· Page 7, line 220: Please consider adding an example/reference showing how water temperature affects the toxicity of NPs.
· Page 7, line 234: Please be specific about the difference between the wild type and the mutant. Which one had higher internalization?
· Page 9, line 300: Please include examples of inhibitors and accelerants.
· Page 9, line 327: Please consider including a numeric number of the common NP concentration ranges in natural aquatic environment.
Author Response
The authors are grateful for the helpful suggestions from the reviewer. With respect to specific requests for clarification, the following revisions have been made.
Response to Reviewer 2 Comments
This is a well put-together manuscript summarizing the effects of nanoparticles on algae. The authors did an excellent job reviewing the current literature of testing nanoparticle toxicity against algae, and concisely yet very informatively outlined the key aspects of NPs and algae interactions, including adsorption and distribution, toxicity mechanisms and outcomes, factors influencing toxicity, as well as the fate of NPs moving up the foot chain through algae. At the end of the review, the authors also pointed out the current knowledge gap and proposed several interesting research areas to further our understanding of NP toxicity. Only minor edits are recommended to improve the overall quality of the manuscript.
Point 1: Page 3, Table 1: In the listed studies, what were the percentage of overall NPs being absorbed by algae cells?
Response 1: Most of the listed studies did not provide the relevant data for adsorption percentage. Only one reference described the percentage of overall NPs absorbed by algae cells and the information has been added in Table 1.
Point 2: Page 3, lines 101-12: Can you add an example figure showing the adsorption kinetics?
Response 2: Thanks for your comment. The example figure of adsorption kinetics from the cited references was not added due to the copyright issues.
Point 3: Figure 1: It would be nice to also include the potential toxic outcomes listed next to the organelles.
Response 3: According to the reviewer’s suggestion, Figure 1 has been revised and more information has been added.
Point 4: Page 5, line 164: Can you list a few examples of the specific ROS produced by exposure to NPs in algae?
Response 4: Examples of the specific ROS produced by exposure to NPs in algae have been added.
Point 5: Page 7, line 220: Please consider adding an example/reference showing how water temperature affects the toxicity of NPs.
Response 5: References have been added in Section 5.1 to show how water temperature affects the toxicity of NPs.
Point 6: Page 7, line 234: Please be specific about the difference between the wild type and the mutant. Which one had higher internalization?
Response 6: The sentence has been rewritten and more information has been added.
Point 7: Page 9, line 300: Please include examples of inhibitors and accelerants.
Response 7: Examples of inhibitor and accelerant were added.
Point 8: Page 9, line 327: Please consider including a numeric number of the common NP concentration ranges in natural aquatic environment.
Response 8: Thanks for your helpful suggestion. To the best of our knowledge, there were currently no available data about accurate NPs concentration range in natural environment (Klaine et al., 2008; Aschberger et al., 2011). However, some data were available for the concentration of NPs in wastewater. C60 fullerenes were detected in wastewater treatment plants in Catalonia (Spain), where their concentrations were at μg/L level and the maximum C60 concentration was 19 μg/L (Farré et al., 2010). An average Ti concentration of 16 μg/L in wastewater effluent was reported by Kiser et al. (2009). Aschberger et al. (2011) summarized the orders of magnitude of modeled and estimated environmental concentrations of several common nanoparticles, where the concentration of nanoparticles in the surface water was generally in the ng/L-μg/L range or even less. Based the above information, the accurate NPs concentration range in natural environment cannot be provided in the paper. But it can be sure that the NP concentration in natural aquatic environment was lower than that in experiment (μg/L-mg/L), since it was at μg/L level even in the contaminated water.
References for Response 8:
Klaine, S.J.; Alvarez, P.J.J.; Batley, G.E.; Fernandes, T.F.; Handy, R.D.; Lyon, D.Y.; Mahendra, S.; McLaughlin, M.J.; Lead, J.R. Nanomaterials in the environment: behavior, fate, bioavailability and effects. Environ. Toxicol. Chem.2008, 27, 1825-1851.
Aschberger, K.; Micheletti, C.; Sokull-Kluttgen, B.; Christensen, F.M. Analysis of currently available data for characterising the risk of engineered nanomaterials to the environment and human health - Lessons learned from four case studies. Environ. Int.2011, 37, 1143-1156.
Farré, M.; Pérez, S.; Gajda-Schrantz, K.; Osorio, V.; Kantiani, L.; Ginebreda, A.; Barceló, D. First determination of C60 and C70 and N-methylfulleropyrrolidine C60 on the suspended material of wastewater effluents by liquid chromatography hybrid quadropole linear ion trap tandem mass spectrometry. J. Hydrol. 2010, 383, 44-51.
Kiser, M.A.; Westerhoff, P.; Benn, T.; Wang, Y.; Perez-Rivera, J.; Hristovski, K. Titanium nanomaterial removal and release from wastewater treatment plants. Environ. Sci. Technol. 2009, 43, 6757-6763.
Note:
1. "Track Changes" function was used during the revision of the manuscript and all the change has been marked in red.
2. More references were cited and all the references have been renumbered.